# MARGINAL DEEP ARCHITECTURES: DEEP LEARNING FOR SMALL AND MIDDLE SCALE APPLICATIONS

**Yuchen Zheng, Guoqiang Zhong & Junyu Dong**
Department of Computer Science and Technology
Ocean University of China
`ouczyc@outlook.com,{gqzhong, dongjunyu}@ouc.edu.cn`

## ABSTRACT

In recent years, many deep architectures have been proposed in different fields. However, to obtain good results, most of the previous deep models need a large number of training data. In this paper, for small and middle scale applications, we propose a novel deep learning framework based on stacked feature learning models. Particularly, we stack marginal Fisher analysis (MFA) layer by layer for the initialization of the deep architecture and call it "Marginal Deep Architectures" (MDA). In the implementation of MDA, the weight matrices of MFA are first learned layer by layer, and then we exploit some deep learning techniques, such as back propagation, dropout and denoising to fine tune the network. To evaluate the effectiveness of MDA, we have compared it with some feature learning methods and deep learning models on 7 small and middle scale real-world applications, including handwritten digits recognition, speech recognition, historical document understanding, image classification, action recognition and so on. Extensive experiments demonstrate that MDA performs not only better than shallow feature learning models, but also state-of-the-art deep learning models in these applications.

## 1 INTRODUCTION

Deep learning methods have achieved desirable performance in many domains, such as image classification and detection, document analysis and recognition, natural language processing, video analysis (Krizhevsky et al., 2012; Chan et al., 2014; Ciresan et al., 2010; Collobert & Weston, 2008; Le et al., 2011). Deep learning methods learn the data representation by using multiple processing layers, which discover the intricate structure of high dimensional data with multiple levels of abstraction (LeCun et al., 2015). For example, for face recognition, the learned features of first layer may be the edges, directions and some local information. The second layer typically detects some object parts which are combination of edges and directions. The higher layers may further abstract the face image by combining the features of previous layers (outline of the eyes, nose, lips). This procedure is very similar with human visual and perceptual system.

In recently years, many deep learning methods have been proposed (l. Boureau & others, 2008; Lee et al., 2009b;a; Hinton & Salakhutdinov, 2006). However, most models meet some difficult problems to solve, such as some parameters need to be randomly initialized, like the weight matrix of two successive layers in deep belief networks (DBNs) and the convolution kernel in convolutional neural networks (CNNs). In addition, traditional deep learning methods need a large scale training data to train the complex networks. It causes many problems in the training process. If we don't initialize the parameters properly, the optimization procedure might need a long training time and fall into local minima. Alternatively, many feature learning models have been proposed to learn the intrinsic structures of high-dimensional data and avoid the curse of dimensionality. In particular, most of them can be trained with small and middle scale of data and their learning algorithms are generally based on closed-form solution or convex optimization. For instance, marginal Fisher analysis (MFA) (Yan et al., 2007; Zhong et al., 2013) is one of the feature learning models that is a supervised method based on the graph embedding framework. It utilizes an intrinsic graph to characterize the intraclass compactness, and another penalty graph to characterize the interclass separability. Its optimal solution can be learned by generalized eigenvalue decomposition. However,

on the one hand, shallow feature learning models cannot work well on the data with highly nonlinear structure; on the other hand, few efforts are made to combine shallow feature learning models for the design of deep architectures.

In order to simultaneously solve the existing problems in deep learning methods and combine the advantages of feature learning models, we proposed a novel deep learning method based on stacked feature learning models. Particularly, we stack marginal Fisher analysis (MFA) layer by layer for the initialization of the deep architecture and call it "Marginal Deep Architectures" (MDA). Firstly, the input data are mapped to higher dimensional space by using random weight matrix. Then we use MFA to learn the lower dimensional representation layer by layer. In the implementation of this architecture, we add some tricks in the training process, such as back propagation, dropout and denoising to fine tune the network. Finally, the softmax layer is connected to the last feature layer. We have compared our MDA with some feature learning methods and deep learning models on different domains of datasets (including handwritten digits recognition, speech recognition, historical document understanding, image classification, action recognition and so on). Extensive experiments demonstrate that MDA performs not only better than shallow feature learning models, but also state-of-the-art deep learning models in small and middle scale applications.

The contributions of this work are highlighted as follows.

1. We propose a novel structure to build a deep architecture. The first hidden layer has twice or quadruple neurons as the input layer. Then we can use some feature learning models layer by layer to learn the compact representations of data. Finally, we set the last layer as a softmax classifier.

2. Traditional deep learning models in general need a large scale training data. Compared with traditional deep learning models, MDA can work better than traditional deep learning models in small and middle scale applications because the initialization of the weight matrices using MFA is much better than that using random initialization.

3. Our MDA can work well in different domains of datasets, such as handwritten digits, spoken letters and natural images. Extensive experiments demonstrate that MDA is a general model to handel small and middle scale data. On the other hand, for large scale datasets, like CIFAR-10, MDA works comparatively with other deep learning methods.

The rest of this paper is organized as follows: In Section 2, we give a brief overview of related work. In Section 3, we present the marginal Fisher analysis (MFA) and the proposed marginal deep architectures (MDA) in detail. The experimental settings and results are reported in Section 4, while Section 5 concludes this paper with remarks and future work.

## 2 RELATED WORK

With the development of deep learning methods, many deep networks have been proposed in recent years (Donahue et al., 2013; Krizhevsky et al., 2012; Long et al., 2015; Zhou et al., 2014). These deep learning models show their powerful performance in various fields, such as image classification and analysis, document analysis and recognition, natural language processing et al. In the area of image analysis, Hinton et al. proposed a large, deep convolutional neural network (Alex net) to classify the 1.2 million high-resolution images in the ImageNet. It uses efficient GPU to speed their method. The results show that a large, deep convolutional neural network is capable of achieving recordbreaking results on a highly challenging dataset using purely supervised learning (Krizhevsky et al., 2012). In order to popularize the deep convolutional neural network, Donahue ea al. proposed DeCAF (Deep Convolutional Activation Feature) which is trained in a fully supervised fashion on a large, fixed set of object recognition tasks (Donahue et al., 2013). DeCAF provides a uniform framework for researchers who can improve and change this framework on some specific tasks. However, its performance at scene recognition has not attained the same level of success. In order to handle this problem, Zhou et al. introduce a new scene-centric database called Places with over 7 million labeled pictures of scenes. Then, they learn the deep features for scene recognition tasks by using the same architecture as ImageNet, and establish new state-of-the-art results on several scene-centric datasets (Zhou et al., 2014). However, these methods based on convolutional operation need very large scale training samples and a long training time. They can not work well on small and middle scale applications.

In other domains, deep learning methods also achieve good performance. Hinton et al. represent the shared views of four research groups that have had recent successes in using DNNs for automatic speech recognition (ASR). The DNNs that contain many layers of nonlinear hidden units and a very large output layer can outperform Gaussian mixture models (GMMs) at acoustic modeling for speech recognition on a variety of data sets (Hinton et al., 2012a). In the area of genetics, Xiong et al. use "deep learning" computer algorithms to derive a computational model that takes as input DNA sequences and applies general rules to predict splicing in human tissues (Xiong et al., 2015). It reveals the genetic origins of disease and how strongly genetic variants affect RNA splicing. In the area of natural language understanding, deep learning models have delivered strong results on topic classification, sentiment analysis et al. Sutskever et al. proposed a general approach, the Long Short-Term Memory (LSTM) architecture which can solve the general sequence to sequence problems better than before (Sutskever et al., 2014). In addition, Hinton et al. proposed autoencoder (AE) networks that is an effective way to learn the low-dimensional codes of high-dimensional data. Based on autoencoder, there are also have many excellent works to handle various tasks. Vincent et al. proposed a denoising autoencoder (DAE) which maked the learned representations robust to partial corruption of the input data (Vincent et al., 2008). The denoising autoencoder which initialize the deep architectures layer by layer is very similar with human visual system. Hinton et al. introduced random 'dropout' to prevent the overfitting which improve many benchmark tasks and obtain new records for speech and object recognition (Hinton et al., 2012b). Then, Vincent et al. proposed stacked denoising autoencoders (SDAE) which based on stacking layers of stacked denoising autoencoders (Vincent et al., 2010). It is very useful to learn the higher level representations and work well on natural images and handwritten digits. However, for the same reason, they also need a large scale training set and a long training time. They have no advantages to handle the small and middle scale applications.

Moreover, in the field of feature learning models, dimensionality reduction plays a crucial role to handle the problems for compressing, visualizing high-dimensional data and avoiding the "curse of dimensionality" (van der Maaten et al., 2009; van der Maaten, 2007). Traditional dimensionality reduction mainly can be classified into three types: linear or nonlinear, like principal components analysis (PCA) (Jolliffe, 2002) and linearity preserving projection (LPP) (Niyogi, 2004) are linear methods, stochastic neighbor embedding (SNE) (Hinton & Roweis, 2002) is a nonlinear method; supervised or unsupervised, such as marginal Fisher analysis (MFA) (Yan et al., 2007; Zhong et al., 2013) and linear discriminant analysis (LDA) (Fisher, 1936) are supervised methods, PCA is an unsupervised method; local or global, like MFA and SNE are local methods, PCA is a global method. Many feature learning models based on geometry theory provide different solutions to the problem of dimensionality reduction. Yan et al. proposed a general formulation about graph embedding framework can exploit new dimensionality reduction algorithms (Yan et al., 2007). If only directly use some feature learning models to extract the good representation from original data, it often eventually couldn't get a good outcome. Considering this situation, we try to choose some excellent feature learning models and combine them with some deep learning algorithms. MFA is one special formulation of the graph embedding models based on this framework. It utilizes an intrinsic graph to characterize the intraclass compactness, and another penalty graph to characterize the interclass separability. Our motivation is to combine the advantage of MFA and deep architectures and propose a new initialization method for deep learning algorithms.

There are also have some excellent works about feature learning models combined the deep architectures (Yuan et al.; George et al., 2014; Ngiam et al., 2011). Yuan et al. proposed an improved multilayer learning model to solve the scene recognition task (Yuan et al.). This model overcome the limitation of shallow, one-layer representations for scene recognition. Trigeorgis et al proposed deep Semi-NMF, that is able to learn such hidden representations from different, unknown attributes of a given dataset (George et al., 2014). Ngiam proposed a deep architectures to learn features over multiple modalities (Ngiam et al., 2011). They showed that multi-modality feature learning is better than one modality and achieved good performance on video and audio datasets. However, in general, we can only obtain data from one modality. In this work, we combine the advantages of MFA and deep architectures, which based on stacked feature learning models (Zheng et al., 2014; 2015), then we use some deep learning tricks, like back propagation, denoising and dropout to fine tuning the network. The advantage of this deep architecture is that we can learn the desirable weight matrix even if the training data is not large enough. And compared with traditional deep learning models and shallow feature learning models, our MDA achieved state-of-the-art results in most cases.

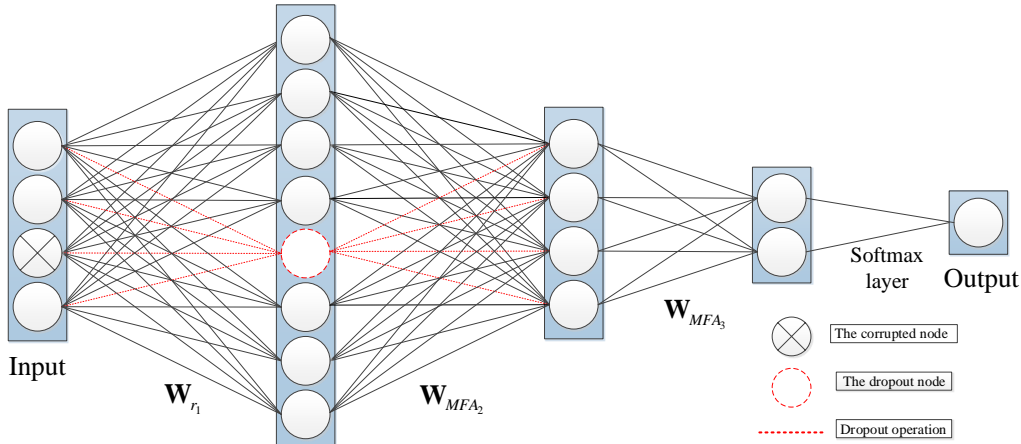

Figure 1: The brief representation for MDA. $\mathbf{W}_{r_1}$ represents the first layer random weight matrix, $\mathbf{W}_{MFA_2}$ and $\mathbf{W}_{MFA_3}$ represent the weight matrixes learned by MFA. The dotted red lines represent the dropout operation, the dotted red circle is the 'dropout' node, and the cross nodes are corrupted. The denoising and dropout operation are completely random. For simplicity, we have omitted bias terms.

## 3 MARGINAL DEEP ARCHITECTURES (MDA)

In this section, we firstly introduce a novel framework of deep architectures, then we introduce marginal Fisher analysis (MFA) and the proposed marginal deep architectures (MDA) in detail. In addition, we also present some deep learning tricks that we used in the MDA model, including back propagation, denoising and dropout.

### 3.1 A NOVEL FRAMEWORK OF DEEP ARCHITECTURES

The feature learning problem is generally formulated as follow. Given $n$ data, $\{\mathbf{x}_1^T, \ldots, \mathbf{x}_n^T\} \in \Re^D$, where $D$ is the dimensionality of the data space, we seeks the compact representations of these data, i.e., $\{\mathbf{y}_1^T, \ldots, \mathbf{y}_n^T\} \in \Re^d$, where $d$ is the dimensionality of the low dimensional embeddings.

In order to improve the accuracy of shallow feature learning models, we use stacked feature learning models to construct the deep architectures (Zheng et al., 2014; 2015), which is a general framework for different applications. In this case, the mapping of data from the original $D$-dimensional space to the resulted $d$-dimensional space can be described as

$$D \Longrightarrow D_1 \Longrightarrow \cdots \Longrightarrow D_i \Longrightarrow \cdots \Longrightarrow D_{p-1} \Longrightarrow d, \tag{1}$$

where $D_1$ is the first higher dimensional space, the number of the node is twice or quadruple as the input layer. $D_i$ represents the dimensionality of the $i$-th intermediate representation space, and $p$ is the total steps of mappings. Here, we can use different feature learning models for the learning of each layer. As the feature learning models are optimized layer by layer, we can obtain the mapping functions between successive layers. The first hidden layer is random by $\mathbf{W}_{r1}$, and the representation is,

$$\mathbf{a}^1 = g(\mathbf{W}_{r_1}^T \mathbf{x} + \mathbf{b}) \tag{2}$$

where, $g(.)$ is a non-linear activation or transfer function. Then, we can use some feature learning models to initialize the next layers. The representations of next hidden layers are,

$$\mathbf{a}^k = g(\mathbf{W}_{F_{k-1}}^T \mathbf{a}^{k-1} + \mathbf{b}) \tag{3}$$

where, $\mathbf{W}_{F_{k-1}}$ is the weight matrix of the $k-1$th layer learned from feature learning models.

### 3.2 MARGINAL FISHER ANALYSIS (MFA)

Based on our novel framework of deep architecture, we introduce Marginal Fisher Analysis (MFA) to build MDA. Here, many traditional feature learning models, such as linear discriminant analysis

(LDA), can be used as building blocks of MDA. Take LDA as an example. It assumes that the data of each class follow a Gaussian distribution. However, this assumption is not often satisfied in the real world. Without this assumption, LDA can not work well to separate the data with nonlinear structure. Alternatively, MFA can solve this problem effectively. Hence, considering the learning capability, we choose MFA as the build blocks of MDA in our work. MFA used the graph embedding framework to set up an intrinsic graph that characterizes the intraclass compactness and another penalty graph which characterizes the interclass separability. The marginal Fisher criterion is defined as

$$\mathbf{W}^* = \underset{\mathbf{W}}{\arg\min} \frac{\text{tr}(\mathbf{W}^T\mathbf{X}(\mathbf{D} - \mathbf{A})\mathbf{X}^T\mathbf{W})}{\text{tr}(\mathbf{W}^T\mathbf{X}(\mathbf{D}^p - \mathbf{A}^p)\mathbf{X}^T\mathbf{W})} \tag{4}$$

where $\mathbf{D}$ and $\mathbf{D}^p$ are diagonal matrices with elements $\mathbf{D}_{ii} = \sum_j \mathbf{A}_{ij}$, and $\mathbf{D}_{ij}^p = \sum_j \mathbf{A}_{ij}^p$, respectively. Then we can learn the projection matrix to multiply PCA's projection and marginal Fisher projection,

$$\mathbf{W}_{MFA} = \mathbf{W}_{PCA}\mathbf{W}^* \tag{5}$$

### 3.3 MARGINAL DEEP ARCHITECTURES (MDA)

In order to combine the advantages of MFA and proposed deep architectures, we propose the marginal deep architectures (or MDA). The MDA inherited from the proposed novel framework of deep architectures is shown in Fig. 1. As an input vector $\mathbf{x} \in [0, 1]^d$, we first map it to higher dimensional space by a random weight matrix $\mathbf{W}_{r1}$. The representation of first hidden layer is computed as

$$\mathbf{a}^1 = s(\mathbf{W}_{r1}^T\mathbf{x} + \mathbf{b}) \tag{6}$$

where, $s(.)$ is the sigmoid function $s(x) = \frac{1}{1+e^{-x}}$, $\mathbf{b}$ is the bias terms, $\mathbf{a}^1$ is the output of first layer. From second layer to $(n-1)$-th layer, we use the weight matrices learned from MFA to map layer by layer.

$$\mathbf{a}^k = s(\mathbf{W}_{MFA_{k-1}}^T\mathbf{a}^{k-1} + \mathbf{b}) \tag{7}$$

The last layer is a softmax regression layer and the number of neuron is the number of category. The cost function is defined as,

$$J(\mathbf{w}) = -\frac{1}{N}\left(\sum_{i=1}^{N}\sum_{j=1}^{K}\mathbf{I}(y_i = j)\log\frac{\exp(\mathbf{w}_j^T\mathbf{a}_i^{n-1})}{\sum_{l=1}^{K}\exp(\mathbf{w}_l^T\mathbf{a}_i^{n-1})}\right) \tag{8}$$

where, $\mathbf{I}(x)$ is the indicator function, $\mathbf{I}(x) = 1$ if $x$ is true, else $\mathbf{I}(x) = 0$. $y_i$ is the label corresponding to $\mathbf{x}_i$. Then the probability that $\mathbf{x}_i$ is classified to $j$ is,

$$p(y_i = j|\mathbf{x}_i, \mathbf{w}) = \frac{\exp(\mathbf{w}_j^T\mathbf{a}_i^{n-1})}{\sum_{l=1}^{K}\exp(\mathbf{w}_l^T\mathbf{a}_i^{n-1})} \tag{9}$$

Taking derivatives, one can show that the gradient is,

$$\nabla J(\mathbf{w}) = -\frac{1}{N}\sum_{i=1}^{N}[\mathbf{x}_i(\mathbf{I}(y_i = j) - p(y_i = j|\mathbf{x}_i, \mathbf{w})] \tag{10}$$

If the $n-1$ layer's neurons are more than the last layer, we can continue using MFA to map it. On the contrary, If the $n-1$ layer's neurons are less than last layer, we can randomly initialize the weight matrix between this two layers. Next, in order to improve the MDA, we introduce back propagation, denoising and dropout operation.

### 3.4 BACK PROPAGATION

In order to adjust the network, we use back propagation (Rumelhart et al., 1986) to compute partial derivative and stochastic gradient descent to update the weight matrixes and the bias terms. For each node $i$ in output layer ($n$-th layer), we compute an error term as

$$\delta_i^n = \nabla J(\mathbf{w}) \tag{11}$$

where, $J(\mathbf{w})$ is the cost function computed from Equ.8 and $\nabla J(\mathbf{w})$ computed from Equ.10. For each node $i$ in $(n-1)$-th to second layer, the error term is computed as,

$$\delta_i^k = \left(\sum_{j=1}^{k+1} w_{ji}^k \delta_j^{k+1}\right) s'(z_i^k) \tag{12}$$

The back propagation procedure relies on computing the gradient of an objective function with respect to the weights of a multilayer stacked modules. It starting from the output at the top and end to the input at the bottom.

## 3.5  DENOISING OPERATION

Vincent et al. proposed the denoising autoencoder to improve the robustness of autoencoder (Vincent et al., 2008). It's very similar with the regularization methods and avoids the "overfitting" problem. The basic idea is to corrupt partial input data by the desired proportion of $\nu$ "destruction". for each input $\mathbf{x}$, a fixed number $\nu d$ of components are chosen at random, and their value is forced to 0, while the others are left untouched. The initial input $\mathbf{x}$ to get a partially destroyed version $\tilde{\mathbf{x}}$ by means of a stochastic mapping,

$$\tilde{\mathbf{x}} \sim q_D(\tilde{\mathbf{x}}|\mathbf{x}) \tag{13}$$

where, $q_D(\tilde{\mathbf{x}}|\mathbf{x})$ is the unknown distribution. Then, for a hidden representation $\mathbf{h}$,

$$\mathbf{h} = s(\mathbf{W}^T \tilde{\mathbf{x}} + \mathbf{b}) \tag{14}$$

In our MDA, we use this idea to improve the network, please refer to Fig. 1 to find clear sight. For the input layer, the output of first hidden layer is represented as,

$$\mathbf{a}^2 = s(\mathbf{W}_{r_1}^T \tilde{\mathbf{x}} + \mathbf{b}_1) \tag{15}$$

where, $\mathbf{W}_{r_1}$ is the first layer random weight matrix, $\mathbf{b}_1$ is the bias term of first layer. The "denoising" operation is established to a hypothetical additional specific criterion: robustness to partial destruction of the input, which means a good intermediate representation is learned from unknown distribution of its observed input. This operation helps for learning more stable structure and avoids the overfitting problem in most cases.

## 3.6  DROPOUT

As the same reason with denoising operation, dropout is a trick to prevent overfitting (Hinton et al., 2012b). When a large feedforward neural network is trained on a small training set, dropout performed well on test set. In order to prevent the complex co-adaptations on the training data, the basic idea of dropout is that each hidden node is randomly omitted from the network with a probability of $\beta$, so a hidden node can't rely on other hidden node. In another view, dropout is as a very efficient way of performing model averaging with neural networks. On test set, we train many separate networks and then to apply each of these networks to the test data. Dropout operation can save the train time and then we average the predictions produced by a very large number of different networks. Fig. 1 shows the dropout operation in our MDA.

## 4  EXPERIMENTS

### 4.1  DATESET DESCRIPTIONS

We evaluate the performance of MDA on five benchmark data sets. The detail of the data is showed in Tab 1. The **USPS** [1] data set is a handwritten digits image data set includes 7291 training samples and 2007 test samples from 10 classes with 256 dimensional features. This task is to recognize the digits 0 to 9. The **Isolet** [2] data set is a collection of audio feature vectors of spoken letters from the English alphabet. It includes 6238 training samples and 1559 test samples from 26 classes with 614 dimensional features. The task is to identify which letter is spoken based on the recorded

---

[1]http://www.gaussianprocess.org/gpml/data/
[2]http://archive.ics.uci.edu/ml/datasets/ISOLET

Table 1: Characteristics of datasets used in evaluation.

| DATASET STATISTICS | | | | | |
|---|---|---|---|---|---|
| dataset | $n$ | train | test | $|\mathcal{Y}|$ | $d$(target) |
| USPS | 9298 | 7291 | 2007 | 10 | 256(32) |
| Isolet | 7797 | 6238 | 1559 | 26 | 614(308) |
| Sensor | 58509 | 46816 | 11693 | 11 | 48(24) |
| Covertype | 581012 | 15120 | 565892 | 7 | 54(27) |
| Ibnsina | 20668 | 17543 | 3125 | 174 | 200(100) |
| CIFAR-10 | 60000 | 50000 | 10000 | 10 | 3072(64) |
| CMU | 49 | 44 | 5 | 3 | 93(24) |

(and pre-processed) audio signal. **Sensor** [3] is a sensorless drive diagnosis data set includes 46816 training samples and 11693 test samples from 11 classes with 48 dimensional features. The features are extracted from electric current drive signals. The task is to classify 11 different classes with different conditions of the drive which has intact and defective components. **Covertype** [4] contains geological and map-based data from four wilderness areas located in the Roosevelt National Forest of northern Colorado. It includes 15120 training samples and 565892 test samples from 7 classes with 54 dimensional features. The task is to identify forest cover type from cartographic variables. For the **IbnSina** [5] ancient Arabic document data set, we use 50 pages of the manuscript for training (17543 training samples) and 10 pages for testing (3125 test samples). The data samples belong to 174 classes of subwords and are of dimensionality 200.

In addition, we also use a large scale dataset **CIFAR-10** [6] to test our MDA on large scale applications. The CIFAR-10 dataset consists of 60000 $32 \times 32$ colour images in 10 classes, with 6000 images per class. There are 50000 training images and 10000 test images. We also test our MDA on a specific task which use the **CMU** motion capture (CMU mocap) data set [7]. The CMU mocap data set includes three categories, namely, jumping, running and walking. We choose 49 video sequences from four subjects. For each sequence, the features are generated using Lawrences method [8], with dimensionality 93 (Zhong et al., 2010). By reason of the few samples of CMU, we adopt 10-fold cross-validation in our experiments and use the average error rate and standard deviation to evaluate the performance.

## 4.2 CLASSIFICATION ON FIVE BENCHMARK DATA SETS

### 4.2.1 BASELINE METHODS

In order to evaluate the performance of MDA, we compared our MDA with 5 deep learning models include autoencoder (AE) (Hinton & Salakhutdinov, 2006), stacked autoencoders, denoising autoencoders (Vincent et al., 2008), stacked denoising autoencoders (Vincent et al., 2010) and stacked denoising autoencoders with dropout, 2 feature learning models, MFA (Zhong et al., 2013; Yan et al., 2007) and PCA (Jolliffe, 2002), PCA deep architecture base on our uniform framework and the classification accuracy on original space.

### 4.2.2 EXPERIMENTAL SETTINGS

All of the deep learning methods have the same settings. The size of minibatch was set to 100, the learning rate and momentum were the default value 1 and 0.5, the number of epoch was set to 400, the dropout rate and denoising rate $\nu$ were set to 0.1. For the AE and SAE, weight penalty of the $L2$ norm was set to $10^{-4}$. For MFA, the number of nearest neighbors for constructing the intrinsic graph was set to 5, while that for constructing the penalty graph was set to 20. The target spaces of MFA and PCA on different data sets were showed in Tab 1. For the USPS data set, The architecture was set to $256 - 512 - 256 - 128 - 64 - 32$. For the Isolet data set ,the architecture was set to $617 - 1324 - 617 - 308$. For the Sensor data set, the architecture was set to $48 - 96 - 48 - 24$.

---

[3] http://archive.ics.uci.edu/ml/datasets/Dataset+for+Sensorless+Drive+Diagnosis#

[4] http://archive.ics.uci.edu/ml/datasets/Covertype

[5] http://www.causality.inf.ethz.ch/al_data/IBN_SINA.html

[6] http://www.cs.toronto.edu/ kriz/cifar.html

[7] http://http://mocap.cs.cmu.edu/

[8] http://is6.cs.man.ac.uk/~neill/mocap/

Table 2: The classification accuracy on different datasets. "ORIG" represents the results obtained in the original data space. 'PDA' represents the PCA deep architecture. 'MDA' represents the MFA deep architecture. The best reslut is highlighted with boldface.

| Method | ORIG | PCA | MFA | AE | SAE | DAE(dropout) | DAE | SDAE | PDA | MDA |
|---|---|---|---|---|---|---|---|---|---|---|
| USPS | 0.8366 | 0.9402 | 0.9392 | 0.9402 | 0.9402 | 0.9581 | 0.9532 | 0.9452 | 0.9586 | **0.9601** |
| Isolet | 0.9467 | 0.9237 | 0.9269 | 0.9519 | 0.9506 | 0.9596 | 0.9519 | 0.9543 | 0.9584 | **0.9622** |
| Sensor | 0.8151 | 0.8042 | 0.8234 | 0.7995 | 0.8325 | 0.8178 | 0.7764 | 0.7870 | **0.8582** | 0.8558 |
| Covertype | 0.5576 | 0.5596 | 0.6057 | 0.7405 | 0.5576 | 0.7093 | 0.7397 | 0.7440 | 0.7458 | **0.7589** |
| Ibnsina | 0.8957 | 0.9190 | 0.9206 | 0.9363 | 0.9184 | 0.9402 | 0.9370 | 0.9261 | 0.9421 | **0.9491** |

Table 3: The structures on 5 data sets. "None" represents without second layer in MDA. "Twice" means the second layer's nodes are as twice as the input layer. "Quadruple" represents the second layer's nodes are as quadruple as the input layer. "Octuple" represents the second layer's nodes are as octuple as the input layer.

| Dataset | None | Twice | Quadruple | Octuple |
|---|---|---|---|---|
| USPS | 256-128-64-32 | 256-512-256-128-64-32 | 256-1024-512-256-128-64-32 | 256-2048-1024-512-256-128-64-32 |
| Isolet | 617-308 | 617-1324-617-308 | 617-2648-1324-617-308 | 617-5296-2648-1324-617-308 |
| Sensor | 48-24 | 48-96-24 | 48-192-96-24 | 48-384-192-96-24 |
| Covertype | 54-27 | 54-108-27 | 54-216-108-54-27 | 54-432-216-108-54-27 |
| Ibnsina | 200-100 | 200-400-200-100 | 200-800-400-200-100 | 200-1600-800-400-200-100 |

For the Covertype data set, we set the architecture to $54 - 216 - 108 - 54 - 27$. Finally, for Ibnsina data set, the architecture was set to $200 - 400 - 200 - 100$.

### 4.2.3 CLASSIFICATION RESULTS

The experimental results are shown in Tab. 2. We can see that our MDA achieves the best results on four dataset except the Sensor dataset, but MDA achieves the second best result on Sensor data set and only below the PDA. The PDA achieves the best result on Sensor data set and the second best results on other data sets. These results demonstrate that our uniform deep architectures achieve the good performance in most case. In addition, MDA not only outperform the traditional deep learning models, but also the shallow feature learning models. It shows that our deep architectures based on stacked some feature learning models can learn the better feature than shallow feature learning models.

### 4.3 EVALUATION

### 4.3.1 DIFFERENT STRUCTURES FOR MDA

In order to evaluate the desired structures of MDA, we changed the node's number of the second layer. For USPS data set, we get rid of the second layer and the architecture was $256-128-64-32$. Then, we set the number of node of the second layer was as twice as the input layer, the architecture was $256 - 128 - 64 - 32$. Next, the number of node was as quadruple as the input layer, the architecture was $256 - 1024 - 512 - 256 - 128 - 64 - 32$. Finally, the node's number is as octuple as the input layer, the architecture was $256 - 2048 - 1024 - 512 - 256 - 128 - 64 - 32$. The structures of other data sets are shown in Tab. 3.

The experimental results are shown in Tab. 4. When the the number of nodes of the second layer is as twice as the input layer, MDA achieved the minimum classification error on all data sets except the Covertype data set. When the number of nodes of the second layer is as quadruple as the input

Table 4: The classification error with different structures on 5 data sets. The best results (minimum error) are highlighted with boldface.

| Dataset | None | Twice | Quadruple | Octuple |
|---|---|---|---|---|
| USPS | 0.0463 | **0.0399** | 0.0433 | 0.0453 |
| Isolet | 0.0398 | **0.0378** | 0.0417 | 0.0430 |
| Sensor | 0.2172 | **0.1442** | 0.1559 | 0.7856 |
| Covertype | 0.3876 | 0.3878 | **0.2411** | 0.3806 |
| Ibnsina | 0.0643 | **0.0509** | 0.0614 | 0.0858 |

Table 5: The classification error on 5 datasets with different number of hidden layers.

| The number of hidden layers | 1 | 2 | 3 | 4 | 5 | 6 | 7 |
|---|---|---|---|---|---|---|---|
| USPS | 0.05780 | 0.05032 | 0.05082 | 0.05182 | **0.03990** | 0.05730 | 0.05132 |
| Isolet | 0.03977 | 0.04169 | **0.03785** | 0.03849 | 0.05452 | 0.04234 | 0.04683 |
| Covertype | 0.27171 | 0.25601 | **0.24110** | 0.26731 | 0.27491 | — | — |
| Sensor | 0.20457 | 0.15522 | **0.14420** | 0.17027 | 0.15567 | — | — |
| Ibnsina | 0.05696 | 0.05184 | **0.05088** | 0.06016 | 0.06720 | — | — |

Table 6: The classification error on gray-CIFAR10 and CMU mocap data sets.

(a) gray-CIFAR10

| Method | Error |
|---|---|
| AE | 0.5117 |
| SAE | 0.5252 |
| DAE(dropout) | 0.5090 |
| DAE | 0.5176 |
| SDAE | 0.5113 |
| PDA | **0.5085** |
| MDA | **0.4947** |

(b) CMU mocap

| Method | Error |
|---|---|
| AE | $0.3970 \pm 0.1343$ |
| SAE | $0.4106 \pm 0.1648$ |
| DAE(dropout) | $0.3970 \pm 0.1343$ |
| DAE | $0.4061 \pm 0.1540$ |
| SDAE | $0.3970 \pm 0.1343$ |
| PDA | $\mathbf{0.3591 \pm 0.0815}$ |
| MDA | $\mathbf{0.3636 \pm 0.0958}$ |

layer, MDA get the worst result on Covertype data set. We can conclude that MDA can work well when the number of nodes of the second layer is as twice or quadruple as the input layer.

### 4.3.2 DIFFERENT NUMBER OF HIDDEN LAYERS FOR MDA

In order to evaluate how many hidden layers adapt to different datasets, we designed some experiments which have different number of hidden layers. We used $1 \sim 7$ hidden layers on USPS and Isolet datasets and $1 \sim 5$ hidden layers on Covertype, Sensor and Ibnsina datasets. The experimental settings were same as previous experiments.

Tab. 5 shows the classification error on 5 datasets with different hidden layers. All the datasets achieved the best results when hidden layer's number is 3 except USPS dataset. The USPS dataset achieved the best result when hidden layer's number is 5. As $1 \sim 3$ hidden layers, with the increase of the number of layers, the classification error is decreasing on all datasets. As small and middle scale applications, we don't need very deep architectures to handle it. As large scale applications, we can design deeper architectures to achieve better performance.

### 4.4 CLASSIFICATION ON LARGE SCALE DATASET CIFAR-10

The previous section introduced the advantages of MDA on small and middle scale applications. In order to evaluate the universality of MDA, we chose a relatively large scale dataset CIFAR-10 to test the performance of MDA.

In our experiments, we first transformed the color images to gray images in order to reduce the dimensionality of input. Then we took one sample as a 1024 dimensional vector which is the input of our MDA. So, we can call this data set gray-CIFAR10. The architecture was set to $1024 - 2048 - 1024 - 512 - 256 - 128 - 64$, the minibatch's size was set to 100, the dropout ratio and denoising ratio were set to 0.1, the number of epoch was set to 400, the learning rate was set to 1, the momentum was set to 0.5. We compared our MDA with previous 6 methods.

Table. 6(a) shows the classification error on gray-CIFAR10, we can see that PDA and MDA achieved the best results in these 7 methods. However, all of the methods on this framework didn't perform well because we use the gray operation.

### 4.5 CLASSIFICATION ON CMU MOCAP DATA SET

CMU mocap data set is a very small dataset that only has 49 samples. Traditional deep learning methods didn't work well in these kind of applications. We test our MDA and PDA and compared them with other 5 deep learning models. The architectures for all deep models (except the PDA) were set to $93 - 186 - 93 - 47 - 24$. Specially, since the CMU mocap data set only has 49 samples, the PCA method only reduce the dimensionality to 49 at most, so the architecture of PDA was set to

$93 - 186 - 24$. The denoising ratio and dropout ratio were set to 0.1 on DAE, DAE with dropout, SDAE, SAE, PDA and MDA. The weight penalty on AE was set to $10^{-4}$. The learning rate was set to 0.01, the momentum was set to 0.5 and the number of epoch is set to 600. The experiment was test on 10-fold cross validation. The experimental results are shown in Tab. 6(b).

In Tab. 6(b), our PDA and MDA achieved the best results in this dataset and have lower standard deviation than other deep learning models. It demonstrates that our PDA and MDA are more stable than other deep learning models. The traditional autoencoder, SDAE, DAE with dropout achieved the same result in this dataset and better than SAE and DAE.

## 5 CONCLUSION

In this paper, we proposed a novel deep learning framework that based on stacked some feature learning models to handle small or middle data sets. Then we introduce MFA in this framework, called MDA. The deep learning tricks like backpropagation, denoising and dropout operation are applied on MDA to improve its performance. Extensive experiments on 7 different type data sets demonstrate that MDA performs not only better than shallow feature learning models, but also state-of-the-art deep learning models on small and middle scale applications. The evaluation of MDA show that how to adjust the parameters make the MDA work well. For future work, we plan to try other feature learning models and explore the different structures for this novel deep learning model. In addition, we plan to explore new deep architectures based on this framework to handle the large scale datasets.

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
