# Peer review of "Marginal Deep Architectures: Deep learning for Small and Middle Scale Applications"

_ICLR 2017 — rejected_

[Official Review · AnonReviewer1 · rating 4 · confidence 4 · 19 Dec 2016]
**Interesting motivation but fail to justify the proposed method**

The authors pointed out some limitations of existing deep architectures, in particular hard to optimize on small or mid size datasets, and proposed to stack marginal fisher analysis (MFA) to build deep models. The proposed method is tested on several small to mid size datasets and compared with several feature learning methods. The authors also applied some existing techniques in deep learning, such as backprop, denoising and dropout to improve performance. 

The new contribution of the paper is limited. MFA has long been proposed. The authors fail to theoretically or empirically justify the stacking of MFAs. The authors did not include any deep architectures that requires backprop over multiple layers in the comparison, which the authors set out to address, instead all the methods compared were learned layer by layer. Will a randomly initialized deep model such as DBN or CNN perform poorly on these datasets? It is also not clear how the authors came up with each particular model architecture and hyper-parameters used in the different datasets. The writing of the paper needs to be significantly improved. A lot of details were omitted, for example, how is dropout applied in the MFA.

[Official Review · AnonReviewer2 · rating 4 · confidence 4 · 19 Dec 2016]
**A deep learning model is proposed where layerwise stacking of a new type of layer (Marginal Fisher Analysis) is used.**

This paper proposes to initialize the weights of a deep neural network layer-wise with a marginal Fisher analysis model, making use of potentially the similarity metric.
 
Pros: 
There are a lot of experiments, albeit small datasets, that the authors tested their proposed method on.

Cons:
lacking baseline such as discriminatively trained convolutional network on standard dataset such as CIFAR-10.
It is also unclear how costly in computation to compute the association matrix A in equation 4.

This is an OK paper, where a new idea is proposed, and combined with other existing ideas such as greedy-layerwise stacking, dropout, and denoising auto-encoders.
However, there have been many papers with similar ideas perhaps 3-5 years ago, e.g. SPCANet. 

Therefore, the main novelty is the use of marginal Fisher Analysis as a new layer. This would be ok, but the baselines to demonstrate that this approach works better is missing. In particular, I'd like to see a conv net or fully connected net trained from scratch with good initialization would do at these problems.

To improve the paper, the authors should try to demonstrate without doubt that initializing layers with MFA is better than just random weight matrices.

[Official Review · AnonReviewer3 · rating 3 · confidence 4 · 21 Dec 2016]

The proposed approach consists in a greedy layer wise initialization strategy for a deep MLP model, which is followed by global gradient-descent with dropout for fine-tuning. The initialization strategy uses a first randomly initialized sigmoid layer for dimensionality expansion followed by 2 sigmoid layers whose weights are initialized by Marginal Fisher Analysis (MFA) which learns a linear dimensionality reduction based on a neighborhood graph constructed using class label information (i.e. supervised dimensionality reduction). Output layer is a standard softmax layer.

The approach is thus to be added to a growing list of heuristic layer-wise initialization schemes.
The particular choice of initialization strategy, while reasonable, is not sufficiently well motivated in the paper relative to alternatives, and thus feels rather arbitrary.
The paper lacks clarity in the description of the approach:  MFA is poorly explained with undefined notations (in Eq. 4, what is A? It has not been properly defined); the precise use of alluded denoising in the model is also unclear (is there really training of an additional denoting objective, or just input corruption?).

The question of the (arguably mild) inconsistency of applying a linear dimensionality reduction algorithm, that is trained without any sigmoid, and then passing its learned representation through a sigmoid is not even raised. This, in addition to the fact that sigmoid hidden layers are no longer commonly used (why did you not also consider using RELUs?).

More importantly I suspect methodological problems with the experimental comparisons: the paper mentions using *default* values for learning-rate and momentum, and having (arbitrarily?) fixed epoch to 400 (no early stopping?) and L2 regularization to 1e-4 for some models. 
*All* hyper parameters should always be properly hyper-optimized using a validation set (or cross-validation) including early-stopping, and this separately for each model under comparison (ideally also including layer sizes). This is all the more important since you are considering smallish datasets, so that the various initialization strategies act mainly as different indirect regularization schemes: they thus need to be carefully tuned. This casts serious doubts as to the amount of hyper-parameter tuning (close to none?) that went into training the alternative models used for comparison. 

The Marginal Fisher Analysis dimensionality reduction initialization strategy may well offer advantages, but as it currently stands this paper doesn’t yet make a sufficiently convincing case for it, nor provide useful insights into the nature of the expected advantages.

I would also suggest, for image inputs such as CIFAR10, to use the qualitative tool of showing the filters (back projected to input space) learned by the different initialization schemes under consideration, as this could help visually gain insight as to what sets methods apart.

[Final Decision · Program Chairs · 06 Feb 2017]
**ICLR committee final decision**

The reviewers unanimously recommend rejection.